# X-ray Fluorescence Techniques in Determining the Habitat Preferences of Species—*Ulva pilifera* (Ulvales, Chlorophyta) from Montenegro Case Study

**DOI:** 10.3390/molecules25215022

**Published:** 2020-10-29

**Authors:** Joanna Czerwik-Marcinkowska, Renata Piwowarczyk, Bohuslav Uher, Ewa Tomal, Anna Wojciechowska

**Affiliations:** 1Institute of Biology, Jan Kochanowski University, 25-406 Kielce, Poland; renata.piwowarczyk@ujk.edu.pl; 2Department of Microbiology, University of Vienna, 1010 Vienna, Austria; uherius@gmail.com; 3Provincial Sanitary and Epidemiological Station, 25-734 Kielce, Poland; ewa.tomal@wsse-kielce.pl; 4Department of Geobotany and Landscape Planning, Nicolaus Copernicus University, 87-100 Toruń, Poland; ankawoj@umk.pl

**Keywords:** elemental composition, macroalgae, Ulvales, WDXRF, TXRF

## Abstract

The paper presents four new sites where bright green *Ulva* thalli were found inhabiting freshwater (a river, a ditch, the Milet Canal) and marine (on the rocky shore of the Adriatic Sea) habitats in Montenegro. The aims of this study were to determine, for the first time, whether specimens of *Ulva pilifera* collected in Montenegro are phylogenetically and morphologically the same species as the one occurring in Europe. Using total reflection X-ray fluorescence (TXRF) and wavelength dispersive X-ray fluorescence (WDXRF) techniques it assessed the elemental composition of their thalli and its influence to colonise new habitats. Elements: Al, As, Ba, Br, Ca, Cl, Cr, Cu, Fe, Hf, I, K, Mg, Mn, Na, Ni, P, Pb, Rb, S, Si, Sr, Ti, V, and Zn were determined. The highest elemental concentrations were found for Ca = 16.3% (using WDXRF) and for Sr = 292 ppm (using TXRF) in the *Ulva* thalli. *Ulva pilifera* analysed from Montenegro, based on classical morphological methods and molecular techniques, are closely related to the same species from inland and coastal waters throughout Europe. The analysis of trace elements showed that the metal content in *Ulva* thalli is correlated with the trace elements in water and sediments. *Ulva pilifera* fits numerous features that make it one of the bioindicators of marine pollution, thanks to its worldwide distribution and capacity to accumulate trace elements.

## 1. Introduction

*Ulva* species grow attached to different substrates (such as soil, rocks, tree bark, leaves, and others macroalgae), but mature thalli are easily detached and become free living algae [1,2]. Due to the fact that *Ulva* species belong to the cosmopolitan genus, occurring in marine (with salinities ranging from 0.5–34%) and freshwater ecosystems [2,3,4] the seasonal massive growth (green tides worldwide) of the free living species is usually a result of high waters eutrophication [5,6]. The genus *Ulva* is one of the first marine genera described by Linneaus (previously known as *Enteromorpha*). Initially *Ulva* and *Enteromorpha* were considered as separate genera, but based on the internal transcribed spacer (ITS) region of nuclear ribosomal DNA (nrDNA) analysis reported by Hayden et al. [7], it was concluded that the genus *Enteromorpha* is synonymous with the genus *Ulva* [8]. It is generally known that the *Ulva* species are difficult to identify morphologically because of its high intraspecific variation and phenotypic plasticity [9,10]. In some habitats (e.g., littoral zone, coral reefs) some species of *Ulva* (e.g., *U. flexuosa*, *U. reticulata*) can be regarded as invasive with fouling potential [11]. However, Mareš et al. [12] showed that marine *Ulva* cannot be recognised as alien or invasive species in inland European waters, but they can be recognised as the primary algae colonisers and indicators of coastal waters eutrophication. Lougheed and Stevenson [13] suggested that *Ulva* species colonise habitats where ecological disturbances are present or conditions of water quality change because of various kinds of contamination.

*Ulva pilifera* (Kützing) Škaloud and Leliaert [14] is currently regarded as a synonym of *Enteromorpha flexuosa* subsp. *pilifera* (Kütz.) Bliding [15]; basionym *Enteromorpha pilifera* Kütz. [16]; the synonym of *Ulva flexuosa* subsp. *pilifera* (Kütz.) M. J. Wynne [17] is regularly encountered in freshwaters, but is equally well represented in brackish waters. In Europe, this species was reported from two distinct areas: a salt marsh in the Czech Republic [12] and from ponds and pools with increased salinity in Poland [18,19], Sweden [15], Slovakia [20], Germany [21], Hungary [12], and the United Kingdom [1]. *U. pilifera* has not until now been observed in Montenegro, although, Kapetanović et al. [22] and Kosanić et al. [23] found two marine ulvalean species *U. lactuca* and *U. intestinalis* in the Adriatic Sea.

This paper presents the study of freshwater and marine *Ulva pilifera* specimens from Montenegro, using morphological, molecular and X-ray fluorescence techniques (total reflection X-ray fluorescence (TXRF) and wavelength dispersive X-ray fluorescence (WDXRF)). It was assumed that *Ulva* marine species adapt effectively and rapidly to develop in freshwater habitats and along the coast of the Adriatic Sea, and also that the *Ulva pilifera* could be used as bioindicator of trace elemental composition in water and sediments in future.

## 2. Results and Discussion

*Ulva* specimens grow in the form of abundant tubular monostromatic thalli in waters (mesotrophic to eutrophic), which are characterised by high levels of nutrients, conductivity (400–1000 μS m^−1^), pH (5.5–9.5), and low oxygen concentration (2.1 mg L^−1^). *Ulva* has been collected in sites rich in N-NH_4_ (0.55 mg L^−1^), P-PO_4_ (0.28 mg L^−1^), and displayed high value conductivity (857.4 μS m^−1^). High concentrations of total iron (0.32 mg L^−1^) and chloride (92.3 mg L^−1^) were defined, in well-oxygenated sites. NaCl formed the major dissolved salt with sulphate and Mg/Ca bicarbonate. High concentrations of sulphate (38.7 mg L^−1^) were noted in habitats where the *Ulva* specimens were found.

Based on morphology, molecular, and elemental analysis (the X-ray fluorescence techniques: TXRF and WDXRF), all of the specimens found at the four sites in Montenegro (Figure 1 and Figure 2) were identified as *U. pilifera*.

### 2.1. Habitat and Morphological Observations

The mature thalli of the studied *Ulva* were irregular in shape, bright to dark green in colour, forming masses of perforated blades, branched and with numerous uniseriate proliferations, and a typical globular cell on the apex (Figure 3). In site B, at a rocky cliff on large rocks washed by the waters of the Adriatic Sea (marine site), square or rectangular apex cells with rounded angles were found. In the apical and central parts of the thalli, in the marine site (B), irregular cell arrangements were observed, whereas in the freshwater sites (A, C, and D) the cells were arranged regularly in longitudinal rows (Figure 4). Considering cells measurements, all material was divided into young and mature thalli collected directly during the studies and analysed separately (there was no culturing in the lab). The free floating thalli were always measured within the stipe region. Thalli of the submerged specimens (young thalli) were from 10.3 cm to 63.6 cm long and 0.3 cm to 1.8 cm wide. Free floating thalli (mature and dying) were from 15.7 cm to 52.3 cm long and 0.6 cm to 2.1 cm wide. Vegetative cells contained one parietal, perforated chloroplast (usually covering the majority of the cell) with 2 or 4 spherical pyrenoids (in the upper basal region). In young thalli, rectangular cells were ranging from 32–56 μm in width to 20–35 μm in length and they formed distinct longitudinal rows. In mature thalli, the vegetative cells were less regular in shape and measured from 24–46 μm in width to 18–32 μm in length, and the rows were not as distinct as those in young thalli. Single row proliferation, culminating in a conical or rounded apex cell of 6.20–7.40 µm in length and 5.80–7.00 µm in width was observed. The thalli of *U. pilifera* created mats with the largest surface (among all the studied samples) in stagnant water (i.e., in the river, in the ditch, and in the Milet Canal—all from Montenegro). There were no zoospores or gametes observed within any of the specimens.

### 2.2. Molecular Phylogenetic Analysis

Molecular analysis confirmed that all of the freshwater and marine specimens occurring for the four sites from Montenegro were indeed *U. pilifera*. The specimens of the same species *Ulva* from freshwater habitats in Poland [18] were compared with the specimens from the similar habitats in Montenegro. The neighbour joining (NJ) and parsimony analysis (MP), as presented in Figure 5 and Figure 6, were congruent with maximum likelihood (ML), considering bootstrap proportions (BP) from ML, NJ, and MP demonstrated at the nodes. The maximum likelihood tree was presented due to the fact that topologies of obtained maximum likelihood, neighbour joining, and parsimony trees were compatible. All new sequences of *U. pilifera* collected from different habitats in Montenegro constituted a monophyletic group, supported by bootstrap analysis. In a Bayesian tree with mapped branches from the MP and ML were considered all phylogenetic analysis, which created trees with very similar clustering at lower levels, but with differences in higher topology. The SH test and robustness of topology, comparing the ML tree with a tree in monophyletic of *Ulva flexuosa* were performed. The results of the SH test stated that the ML tree was significantly preferable than the tree with monophyletic *U. pilifera*, as well as the, *rbc*L data set was combined with related sequences from GenBank (BLAST). There were no differences in the coded polypeptide amino acid sequences. The cells of *Ulva* thalli (Poland) are not morphologically similar to those from Montenegro (Table 1), but the monophyletic clusters are characterised by unique morphological traits representing synapomorphy, which further allow them to be distinguished from their sister clades.

### 2.3. Elemental Analysis

In *Ulva* thalli 25 elements: Al, As, Ba, Br, Ca, Cl, Cr, Cu, Fe, Hf, I, K, Mg, Mn, Na, Ni, P, Pb, Rb, S, Si, Sr, Ti, V, and Zn, using WDXRF and TXRF techniques, were identified (Table 2 and Table 3). Moreover, 14 elements were classified by WDXRF and 11 elements by TXRF. At site A of the *Ulva pilifera* thalli the calcium (Ca) levels showed the highest concentration, relatively large in site D, while at site B there were only trace amounts using WDXRF technique (see Table 2). At site B, using TXRF technique, the strontium (Sr) reached the highest value, while in sites A and D the amounts were 50% lower (Table 3). The elements concentration using WDXRF technique at the site A for barium (Ba) and phosphorus (P), similarly to the site B for iodine (I) and bromine (Br), and at the site D for bromine (Br) and sodium (Na) were the lowest (see Table 2). While using TXRF technique (Table 3) at sites A and D, the elements concentration for arsenic (As) and chromium (Cr), and at site B for vanadium (V) and lead (Pb) were the lowest.

### 2.4. Comparison of Data Analysis

The Principal Component Analysis (PCA) indicated that the *Ulva* elemental composition from site B (from the Adriatic Sea) differs from that of sites A and D. The analysis was done only from sites A, B and D, while site C was below the limit of significance (and therefore was not included in the analysis). The *Ulva* thalli from site B were characterised by more strontium and zinc, whereas those from A and D had more copper and titanium (Figure 7). The first axis explains 93% of the variation, because the variation in the data was small and the differences between these sites were not too large. The whole group of chemical elements visible on the left side of the graph were no different among the sites. The Canonical Variate Analysis (CVA) showed that *Ulva* from site B (marine) contained more microelements than those from the sites A and D (Figure 8). The microelements variable explains 50% of the data variability. The CVA confirmed the results obtained from the PCA. Apaydin et al. [24] investigated the contents of trace elements by the energy-dispersive X-ray fluorescence spectrometry (EDXRF) in *Ulva lactuca* from eight different regions of Istanbul (Turkey). Their results demonstrated that *Ulva* thalli contain trace element contamination (but no toxic element), and that the mineral content of *Ulva lactuca* varies according to the composition of sediment, sea water, sand, and rock.

### 2.5. Distribution and Habitat Preferences of the Ulva Species

*U. pilifera* is considered a cosmopolitan species that has a worldwide distribution [13]. The distribution of *U. pilifera* was based on all the references available. Table 4 presents the occurrence and habitats preference of *U. pilifera* in most continents, excluding Antarctica.

The taxa of the genus *Ulva* were difficult in terms of morphological identification, due to the plasticity of the thallus and cellular details [1,41]. *Ulva pilifera* is commonly encountered in both freshwaters, and slightly saline waters [42,43]. Mareš et al. [12] stated that *Ulva flexuosa* subsp. *paradoxa* and *Ulva pilifera* are freshwater algae which dominate in Europe habitats, and this is confirmed by the morphological and molecular results. Kapetanović et al. [22] and Kosanić et al. [23] found *Ulva lactuca* and *Enteromorpha intestinalis* in Montenegro on the Adriatic Coast (in waters with a high salt content). In studies *Ulva pilifera* was observed in one marine and in three freshwater sites. Variable environmental conditions are responsible for the plasticity of Ulvales morphology characters [44]. On the one hand, pyrenoids number is believed by some authors to be of taxonomic values but it is questioned by others [45]. *Ulva pilifera* found in the four sites from Montenegro formed longer, wider thalli and thicker cell walls than the same species observed in Poland and the Czech Republic. Morphological analysis of this species from 21 freshwater sites in Poland [41] displayed tubular, curled and bubbled thalli with strongly corrugated surfaces and uniseriate branches (4.7–40.4 cm in length, 0.10–23.0 mm in width) containing from 1 to 4 pyrenoids. Whereas the *Ulva pilifera* found by Mareš et al. [12] in enriched freshwaters habitat formed tube-like or leaf-like thalli, free floating masses, abundant or absent branching from 7 to 18 (25) µm in length, with from 2 to 4 pyrenoids.

Poole and Raven [46] and Rybak et al. [41] stated that the subspecies of *Ulva flexuosa* is characterised by wide tolerances to factors such as water salinity, temperature and nutrients. However, *Ulva’s* are known to grow in a wide range of tolerance to salinity, but optimal salinity is different for adult thalli or spores [47]. The local conditions of *Ulva* habitats depend also on the tolerance range of salinity for individual species. Rybak et al. [41] found that *Ulva pilifera* in different freshwater ecosystems (natural and anthropogenic) dominated most often lakes, water basins, and rivers, where the salinity did not exceed (i.e., 270 mg L^−1^ Cl^−^).

Both eutrophication and climatic factors, such as light and temperature, promotes the proliferation and growth of the green tide forming algae known as *Ulva* species. Factors, such as temperature, pH, salinity, and nutrient availability, have limited spatial and temporal distribution of *Ulva pilifera* in different habitats. Holzinger et al. [48] observed that algae respond rapidly to changes in water quality due to different environmental factors and that *Ulva compressa*, characterised by *rbc*L phylogeny, is a common species in the Mediterranean Sea. *Ulva* as an intertidal species tolerates repeated desiccation-rehydration cycles in nature, with drastic variation in salinity requiring a high osmotic potential and water-holding capacity [49]. West and Pitman [50] studied ionic relations in *Ulva,* and they found an accumulation mechanism for potassium, while sodium accumulation was excluded. However, Holzinger et al. [48] concluded that structural changes in the cell walls of *Ulva compressa* during desiccation-rehydration cycles play a key role in surviving high and low tides. *Ulva pilifera* existing both in the freshwater and marine environment, implies high ecological plasticity. Typically, *Ulva* species are regarded as good indicators of heavy metal concentration in sea water [51]. The concentration of many elements (including heavy metals) in algae fluctuates with the seasons due to changes in their bioaccumulation during development and independent of ion concentration in the environment [52]. According to Rybak et al. [41], the highest metal concentrations were for calcium (Ca) and magnesium (Mg) in the thalli of freshwater *Ulva*, both in lake and the river populations. In the Montenegro specimens, the highest elemental concentrations were for calcium (Ca) (freshwater site A) and strontium (Sr) (sea water site B), which has physical and chemical properties similar to calcium (Ca). Calcium and barium have physical and chemical properties similar to strontium, and all three of them are vertical neighbours in the periodic table.

Montenegro’s diversity of geology, landscapes, climate types and soils, also different water habitats and geographical location on the Balkan Peninsula and Adriatic Sea, result in remarkable biological diversity. Due to these specific qualities, Montenegro ranks among Europe’s and the world’s most biologically diverse hot-spots. Since 2000, in Montenegro, the National Environmental Monitoring Programme discovered 1500 species of algae (some 1000 taxa are known from Skadar Lake), 8000 species of vascular plants and 372 Balkan endemic taxa, therefore, the number of taxa per area is 0.837 which constitutes the highest index for all known European countries [22]. Future work should extend the search for other new localities of *Ulva* species in the Balkans.

## 3. Materials and Methods

### 3.1. Study Site

Specimens of *Ulva pilifera* were collected (in 50 mL sterile clean bottles) on May 2015 from four sites in Montenegro: (A) 42°23′24″ N, 18°42′51″ E, 5 May 2015, the outskirts of Peninsula Luštica, in the river 2 km SE from Durasevici (Figure 1). The river is overgrown with *Phragmites*, *Callitriche*, and *Zannichellia* communities. There are numerous and scattered salt springs and saline wetlands in the immediate vicinity of the river. (B) 41°55′31″ N, 19°11′58″ E, 6 May 2015, NW of Old Town of Ulcinj city; the site is located at the base of a rocky cliff on large rocks washed by the waters of the Adriatic Sea. (C) 41°54′52″ N, 19°16′34″ E, 6 May 2015, a ditch near Ulcinjska Solana (the vast space in the area of Adriatic Coast, where salt is produced by the evaporation; this is one of the largest salt basins in the Balkans. (D) 41°54′52″ N; 19°16′36″ E, 6 May 2015, the Milet Canal in Donji Štoj near Ulcinjska Solana (Figure 2). Besides *Ulva pilifera*, there were also large quantities of *Potamogeton pectinatus* (L.) Börner. Each sampling sites was recorded by Garmin Differential GPS, and all specimens were photographed fresh to show cell arrangement, cell size, and branching patterns.

### 3.2. Samples Preparation

A total of 30 *Ulva* samples were collected, subsequently, each sample was washed clean with water and thoroughly dried with absorbent paper. Fresh material was pressed and dried in herbarium sheets, whilst part of the material from the *Ulva* specimens was cleaned well using distilled water to remove epiphytes and preserved in silica gel (Sigma Aldrich, St. Louis, MO, USA) for molecular analysis. Each sterile clean bottles was labelled with the date of collected, the habitats number, and sample number. For morphological studies (including measurement of *Ulva* thalli, presence or absence of indentation, cell wall thickness, cell dimensions and number of pyrenoids within the chloroplast), the specimens were examined using a JENAMED 2 microscope. Morphometric diagnostic features were measured and photographed using an inverted Nikon Eclipse Ti microscope, and confocal equipment a Nikon A1 (Nikon, Tokyo, Japan). The 405.488.561 nm laser heads and detection system was a PMT-DU4 detector in the range of 400–820 nm, Plan Apo VC 100XOil inverse lens, Nikon C-HgFiE mercury illumination, and a DS0Fi1C-U3 digital camera. Data acquisition in Nikon NIS-Elements obtained digital images, processed with Adobe Photoshop CS5 (Adobe, San Jose, CA, USA). The elemental analysis of *Ulva* thalli was carried out using wavelength dispersive X-ray fluorescence (WDXRF) and total reflection X-ray fluorescence (TXRF) in the Laboratory of X-ray methods (Institute of Physics, Jan Kochanowski University in Kielce, Poland). The *Ulva* specimens were morphologically identified using Bliding (1963) and following [14,17].

Ulva preparation for the WDXRF technique. For the wavelength dispersive X-ray fluorescence analysis, the samples were prepared as a pellet. Prior to pellet formation, the *Ulva* samples were dried and ground with the compact mill MiniMill2 at the rotation speed 200 rpm and grinding time t = 1 min. Next, the powdered *Ulva* (about 5 g) was mixed with wax binder (C_18_H_36_O_2_N_2_), whose mass was about 5% of the algae mass, and afterwards it was formed into the pellet using pressure equal to 20 tons for 30 s. WDXRF measurement was carried out for 40 min.

Ulva preparation for the TXRF technique. The TXRF analysis requires a liquid form of the sample. For this purpose the powdered *Ulva* samples weighing (~0.03 g) was mixed in an Eppendorf tube with 500 µL of high purity 65% HNO_3_ (Merck). Due to the violent reaction of acid with the *Ulva* sample, the acid was added very carefully in the following stages: 100 µL + 100 µL + 100 µL + 200 µL. Finally, 40 µL of 100 ppm of SeO_2_ (in HNO_3_ 2–3%; Merck) was added to the mixture as internal standard. The solution was next mineralised in a microwave oven. Next, the solution was allowed to stand for 24 h in order for total mineralisation. Then, the 5 µL of sample solution was pipetted into a silicon sample carrier and dried on at hot plate at 60 °C. Finally, each *Ulva* sample was analysed using TXRF technique (the measurement time was 60 min). The concentration of elements obtained in the solution (*Ulva* powder + acid + Se as internal standard) was converted into the dry weight of *Ulva*. Eppendorf tubes, pipette tips and sample carriers were previously washed in 10% nitric acid, rinsed with deionised water and then dried by centrifugation.

### 3.3. Physicochemical Water Quality

Water samples from all the sites were collected using clean hard rubber bottles. Water temperature (°C), pH, specific conductivity (µS cm^−1^) and total dissolved solids (mg L^−1^) were measured in situ using the Bernant 30 pH meter, while dissolved oxygen (DO) was determined using YSI 51B DO meter. Detailed hydrochemical characteristics of each study sites, including irons, nutrients, trace elements and metals, were determined following [53]. The ions: Ca^2+^, Cl^−^, HCO_3_, Mg^2+^, Na^+^ and SO_4_^2−^ were measured using ion-exchange chromatography (ICS 3000 Dionex). The nutrients: NO_2_, NO_3_, NH_4_^+^, TP, and SRP were determined by molecular absorption spectroscopy (UniCam 969 AA Spectrometer).

### 3.4. Molecular and Phylogenetic Analysis

Total genomic DNA was extracted from 100 mg sample of fresh tissue (*Ulva* specimens from Montenegro and Poland, each one separately) and grounded with liquid nitrogen using the DNeasy Plant Mini Kit (Qiagen GmbH, Hilden, Germany). HotStartTaq DNA Polymerase (Qiagen) was used to amplify the selected genomic regions using newly designed ITS primers (ITS-F 5′-CTG CGG AGG GAT CAT TGA AA-3′; ITS-R 5′-GTA GCG AGC TAC CTA CCT AGT-3′) and *rbc*L primers (*rbc*L-F 5′-CAA GGT CCA CCT CAT GGT ATT C-3′; *rbc*L-R 5′-GTC ACC TGA CAT ACG TAA A-3′). Thermal cycling parameters were as follows: initial denaturation at 95 °C for 15 min followed by 40 cycles at 94 °C for 60 s, 58 °C (ITS) or 62 °C (*rbc*L) for 30 s, and 72 °C for 40 s. Direct sequencing of PCR products were done by GenoMed S.A. (Warsaw, Poland), and manually inspected using Chromas Lite 2.1.1 (Technelysium Pty Ltd., South Brisbane, Australia). The divergence (in%), distance in the sequence, have a Montenegro species in ITS and *rbc*L sequences. All others sequences for phylogenetic trees were taken from GenBank and they were highly similar to our sequences. The phylogenetic trees were constructed by the maximum-likelihood algorithm of the MEGA 6 program Tamura et al. [54]. The obtained sequences were deposited in GenBank under the following accession numbers: KX398148 for *rbc*L, KX398149 for ITS (*Ulva* from Montenegro), KX398150 for *rbc*L, KX398151 for ITS (*Ulva* from Poland).

### 3.5. Wavelength Dispersive X-ray Fluorescence (WDXRF) Technique

The spectrometer Axios (PANalytical) with X-ray tube with Rh anode was used to take WDXRF measurements. The X-ray tube has 2.4 kW maximum power and 60 kV maximum voltage, as well as 100 mA maximum current. The primary beam, its intensity, and spectrum was modified using different filters. The excitation of the X-rays sample, early generated in an X-ray tube, is a source of information about elemental composition. Due to the analysis of crystal, according to Bragg’s law, characteristic polychromatic X-ray was dispersed into monochromatic constituents using a beam appearing from a sample surface and collimated by a collimator. Five crystals: LiF200, Ge111, PE 002, PX1, and LiF220 are present in the Axios spectrometer and their choice is necessary to cover the wavelength. Two detectors: a flow counter is used to optimise elements detection up to iron (Fe) or a scintillation counter used for heavier elements served the characteristic X-rays. Elemental analysis from oxygen (O) to uranium (U) was carried out by Axios spectrometer. The measurements of elemental analysis were performed in vacuum.

### 3.6. Total Reflection X-ray Fluorescence (TXRF) Technique

TXRF measurements were performed with the S_2_ PICOFOX spectrometer (Bruker AXS Microanalysis GmbH). On the basis of [55,56] the characteristic X-rays were excited in the samples with the 30 W Mo-anode X-ray tube operated at 50 kV with an electron current of 0.6 mA. Fluorescence X-rays from the samples were detected by Peltier-cooled XFlash silicon drift detector (with an area 10 mm^2^) having an energy resolution of approx. 160 eV at 100 kcps for Mn-Kα line [56]. The measurements were performed in the air. The PICOFOX spectrometer allows measurement of characteristic X-rays of elements from Al (aluminium) to U (uranium), with the exception of zirconium (Zr) to ruthenium (Ru). A good detection value (from 10 ppb in the case of real multielemental sample) is achieved as a result of optimisation of the setup geometry and use of a multilayer monochromator.

### 3.7. Statistical Analysis

Principal Component Analysis (PCA) was used to show similarities and differences in the *Ulva* elemental composition among the studied sites. The ordination diagram presents the arrangement of variables: elements (geometric symbols) and sites (vectors) in the ordination space. The closer vectors location indicates larger chemical elemental similarity on the sites. For the purpose of the performed analysis, the assigned elements were divided into chemical groups. *Ulva* thalli elemental composition in particular sites by Canonical Variate Analysis (CVA) was determined. Three chemical groups of elements were distinguished: heavy metals (Ti, Cr, Ni, As, Hf, and Pb), microelements (Ba, Cl, Fe, Mn, I, V, Cu, Zn, Rb, and Sr), and macroelements (Al, Br, Ca, K, Mg, Na, P, S, and Si), and all of these were encoded accordingly in the system 0–1. The data were correlated with the elemental *Ulva* composition. Monte Carlo permutation test was also performed to determine the statistical significance of the used variables. On the ordination diagram, the sites were marked with triangles and the elemental chemical groups were marked with vectors. Both ordination analyses were carried out in Canoco 5.0 [57].

## 4. Conclusions

A green macroalgae was collected in Montenegro from four sites (freshwater and marine): a river, a ditch, the Milet Canal, and on the rocky shore of the Adriatic Sea. A detailed morphology, autecology, and molecular studies allowed to identify the *Ulva* population as *Ulva pilifera.* WDXRF and TXRF techniques were applied for the analysis of elemental composition of *Ulva* thalli. Molecular studies demonstrated a close phylogenetic relationship between *Ulva* species isolated from Poland, the Czech Republic and Montenegro. It was focused on the correlation between different kinds techniques applied to taxonomic identification of the *Ulva* species and its elemental properties. In elemental analysis, 25 elements were determined, i.e., Al, As, Ba, Br, Ca, Cl, Cr, Cu, Fe, Hf, I, K, Mg, Mn, Na, Ni, P, Pb, Rb, S, Si, Sr, Ti, V, and Zn. The concentrations of the metals Cr, Cu, Ni, PB, and Zn in the marine site were higher than in freshwater sites and significantly correlated with the content of trace elements in water and sediments. CVA and PCA statistical analysis confirming the obtained results were performed. The application of WDXRF and TXRF techniques as chemotaxonomic marker might be useful, in the future for identifying freshwater and marine *Ulva* species and also determining their bioindicative properties.

## Figures and Tables

**Figure 1 molecules-25-05022-f001:**
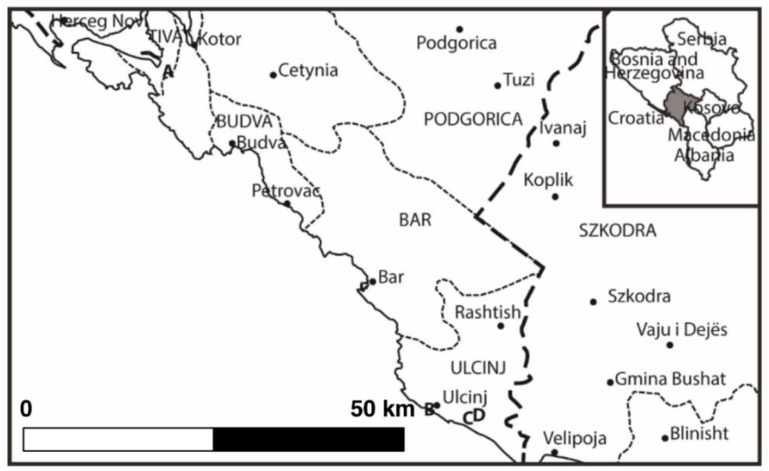
Location of *Ulva pilifera* sampling sites in Montenegro: **A**, the River Ulcinj 2 km SE from Durasevici; **B**, NW of the Ulcinj City, on the Adriatic Coast; **C**, the ditch near Ulcinjska Solana; **D**, the Milet Canal in Donji Štoj near Ulcinjska Solana.

**Figure 2 molecules-25-05022-f002:**
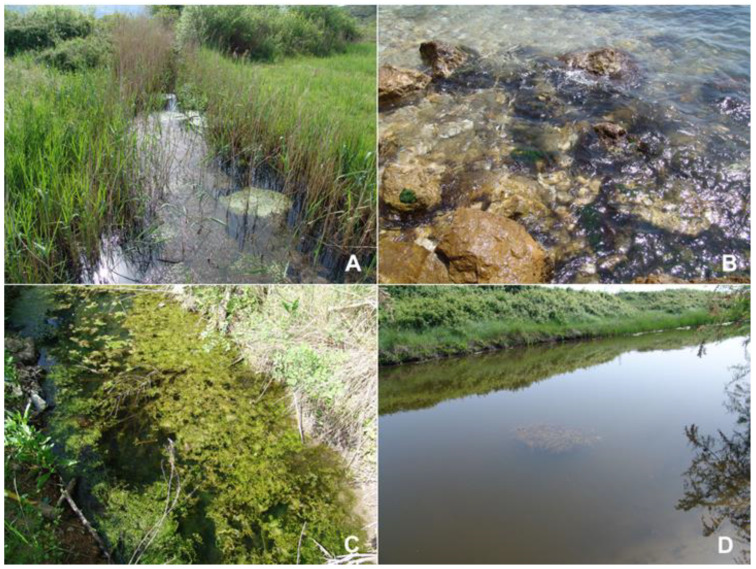
Landscape views showing of the sampling sites of *Ulva pilifera*: (**A**) the river (freshwater), mass growth of the green thallus of *Ulva*; (**B**) a rocky cliff on large rocks of the Adriatic Sea (marine); (**C**) the ditch (freshwater); (**D**) the Milet Canal (freshwater).

**Figure 3 molecules-25-05022-f003:**
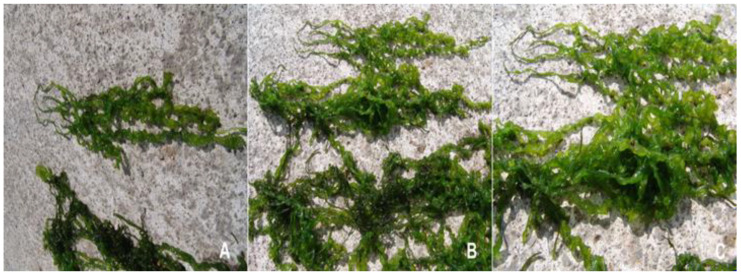
Thalli of *Ulva pilifera* from the river (**A**), and from the Adriatic Sea (**B**,**C**). Thallus corrugated, laminar, light green filiform, and intensive ramification.

**Figure 4 molecules-25-05022-f004:**
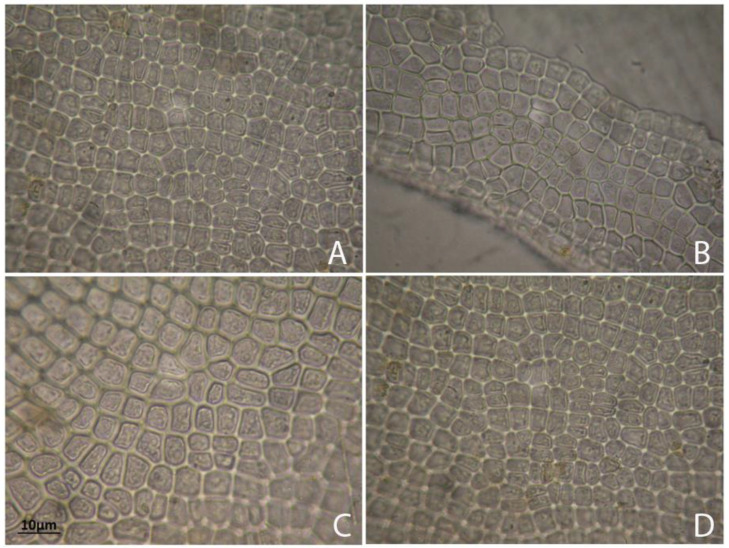
*Ulva pilifera* (**A**,**D**) close-up view on the slightly rounded and/or polygonal vegetative cells well-ordered in long rows in the mid region of the thallus with parietal chloroplast and 2–3 pyrenoids; (**B**) rectangular cells in a branch showing parietal chloroplasts covering the majority of the cell walls; (**C**) irregularly arranged vegetative cells in the thallus with 2–3 pyrenoids. Scale bars: 10 µm.

**Figure 5 molecules-25-05022-f005:**
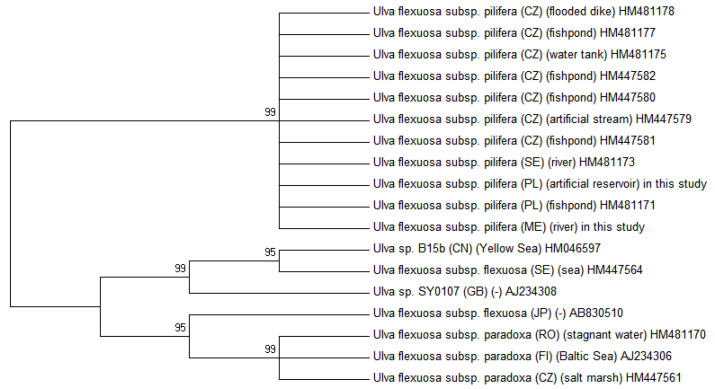
Maximum likelihood of ITS DNA sequences of *U. pilifera* samples obtained from Montenegro and Poland. Numbers on branches bootstrap support analysis 1000/50. The phylogenetic tree was constructed by the ML algorithm of the MEGA 6 program. Location and habitat of *Ulva* species taken from GenBank [3,12].

**Figure 6 molecules-25-05022-f006:**
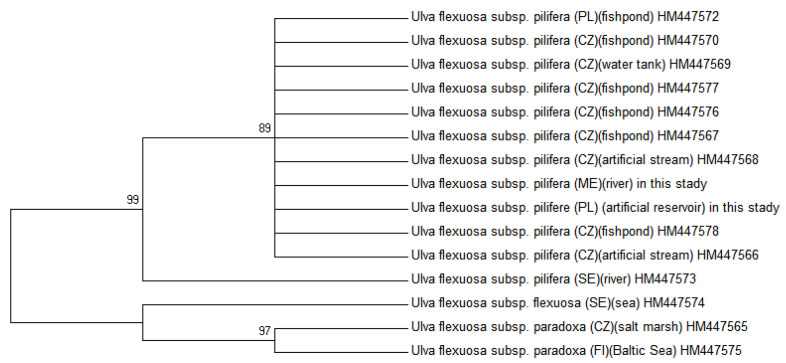
Maximum likelihood tree of *rbc*L DNA sequences *U. pilifera* samples obtained from Montenegro and Poland. Numbers on branches represent bootstrap support analysis 1000/50. The phylogenetic tree was constructed by ML algorithm of the MEGA 6 program. Location and habitat of *Ulva* species taken from Mareš et al. [12].

**Figure 7 molecules-25-05022-f007:**
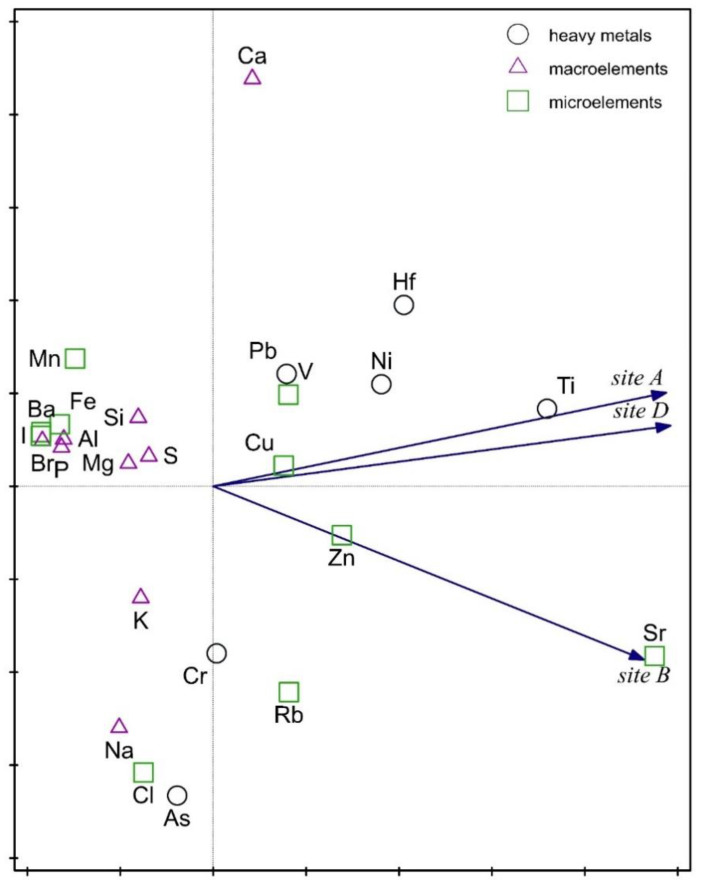
Principal Component Analysis (PCA) showed a composition data of the *Ulva* elements from the studied sites. The first axis explained 93% of the data variation. The vectors indicate the sites and the symbols elements divided into chemical groups.

**Figure 8 molecules-25-05022-f008:**
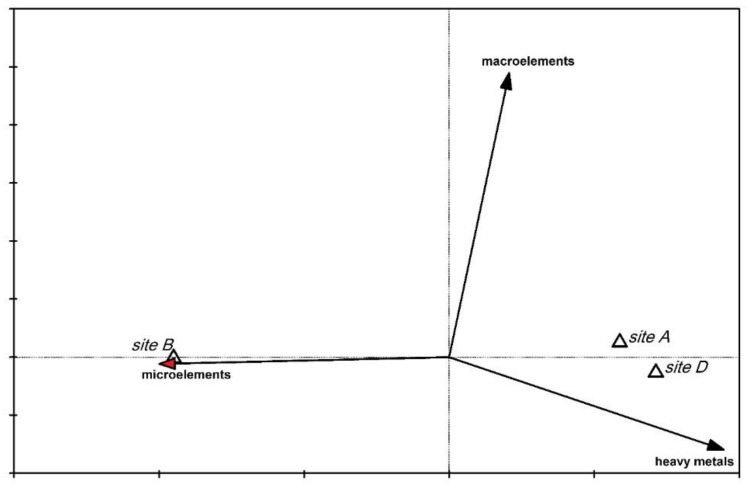
Canonical Variate Analysis (CVA) for *Ulva* elemental composition from study sites with the chemical groups of elements. The sites were marked with triangles and the elemental chemical groups were marked with vectors. The red tip of the vector signifies a statistically significant variable and 50% of the variance of the data.

**Table 1 molecules-25-05022-t001:** Morphological characteristics of *Ulva pilifera* from Montenegro and Poland [18].

Characters	Montenegro	Poland
morphology	tube-like	tubular, curled, and bubbled
thalli colour	bright to dark green (mature yellow-green)	bright green
branching	frequent	abundant to almost absent
uniseriate branches	frequent to rare, obtusely rounded ends	present
cell shape	rounded polygonal to quadrangular	rectangular, polygonal
structure of branch tips	globular cell on the apex	rectangular or square
number of pyrenoids	2–3–4	2–4
chloroplast shape	parietal	parietal, girdle-shaped
cell size: length of cells (μm) width of cells (μm)	42.25–66.56 7.58–19.80	32.05–55.20 10.50–35.50
thallus size: length of thalli (cm) width of thalli (mm)	6.50–56.20 0.20–1.50	10.1–65.1 0.20–2.50
cell arrangement	disordered, in small groups or short rows	disordered, in small groups or short rows
habitat	brackish and marine waters (sea coast)	enriched freshwaters (ponds, pools, small streams)
mode of life	attached or free floating	attached or free floating, occasionally in masses

**Table 2 molecules-25-05022-t002:** Element concentrations (±measurement expanded uncertainty, expanded factor k = 2) determined in *Ulva pilifera* thalli using wavelength dispersive X-ray fluorescence (WDXRF) technique. Concentrations are given in %. The analysis were done only from sites A, B, and D while site C was below the limit of significance.

Element	Concentration (%) ± Expanded Uncertainty		
	Site A	Site B	Site D
**Al**	0.186 ± 0.026	0.243 ± 0.030	0.310 ± 0.034
**Ba**	0.020 ± 0.008	--------	--------
**Br**	--------	0.039 ± 0.012	0.013 ± 0.006
**Ca**	16.300 ± 0.200	0.854 ± 0.056	10.300 ± 0.200
**Cl**	0.560 ± 0.044	7.630 ± 0.170	0.498 ± 0.042
**Fe**	0.120 ± 0.020	0.129 ± 0.022	0.366 ± 0.036
**K**	0.698 ± 0.050	3.330 ± 0.110	1.450 ± 0.070
**Mg**	0.999 ± 0.060	1.390 ± 0.070	1.550 ± 0.060
**Mn**	0.299 ± 0.032	--------	1.020 ± 0.060
**Na**	0.454 ± 0.040	4.990 ± 0.130	0.149 ± 0.024
**P**	0.070 ± 0.016	0.242 ± 0.030	0.372 ± 0.036
**S**	1.860 ± 0.080	1.860 ± 0.080	1.640 ± 0.080
**Si**	1.670 ± 0.080	1.250 ± 0.070	1.640 ± 0.080
**I**	-----------	0.009 ± 0.006	-------

The uncertainty values are the expanded uncertainty with an expansion of level about 95% with factor of k = 2. The value of extended uncertainty is given as a range (±U95%).

**Table 3 molecules-25-05022-t003:** Element concentrations (±measurement expanded uncertainty, expanded factor k = 2) determined in *Ulva pilifera* using total reflection X-ray fluorescence (TXRF) technique. Concentrations are given μg/g. The analysis were done only from sites A, B, and D while site C was below the limit of significance.

Element	Concentration (μg/g) ± Expanded Uncertainty		
	Site A	Site B	Site D
**Ti**	148.000 ± 2.000	82.600 ± 1.000	135.000 ± 2.000
**V**	13.800 ± 0.800	7.140 ± 0.540	8.090 ± 1.080
**Cr**	1.800 ± 0.640	9.190 ± 0.420	4.530 ± 0.900
**Ni**	22.200 ± 0.200	15.700 ± 0.200	39.200 ± 0.400
**Cu**	10.700 ± 0.200	9.030 ± 0.180	7.390 ± 0.240
**Zn**	11.300 ± 0.200	19.400 ± 0.200	20.100 ± 0.200
**As**	0.531 ± 0.138	11.100 ± 0.200	1.900 ± 0.140
**Rb**	5.680 ± 0.140	22.200 ± 0.200	6.920 ± 0.140
**Sr**	292.000 ± 2.000	541.000 ± 2.000	238.000 ± 2.000
**Hf**	58.400 ± 0.400	15.700 ± 0.200	27.200 ± 0.400
**Pb**	9.510 ± 0.180	6.030 ± 0.140	13.300 ± 0.200

The uncertainty values are the expanded uncertainty with an expansion of level about 95% with factor of k = 2. The value of extended uncertainty is given as a range (±U95%).

**Table 4 molecules-25-05022-t004:** Occurrence (in freshwater and marine ecosystems) and habitat preference of *Ulva pilifera*.

Region	Habitats	References	Notes
**Africa**			
Egypt	limno-rheocrenic, thermal, mineral (chloride, sodium, sulphate) spring known as “Ain Abu Sherouf” in the Siwa Oasis; Red Sea coasts; brackish estuaries of the Bile River	Shaaban et al. [25] Saber et al. [4]	the Western Desert of Egypt
**Asia**			
China	Yellow Sea, South China Sea	Wang et al. [26] Phang et al. [27]	the Subei Shoal coastal waters; bordered by Philippines
Japan	freshwater and brackish	Shimada et al. [28]	
Korea	marine	Lee et al. [29]	
Turkey	Aegean Sea	Taskin et al. [30]	
Vietnam	marine	Nguyen et al. [31]	mats
**Australia**			
	marine	Kraft et al. [32] Kirkendale-Saunders and Winberg [33]	algal blooms, southern Australia
**Europe**			
Czech Republic	lakes, ponds, streams, rivers in Central Europe	Mareš et al. [12] Messyasz and Rybak [34]	
Germany	pond	Kopp [20]Mareš et al. [12]	
Great Britain	near the top of the shore, on rocks or other algae, on open coasts or in estuaries and harbours	Brodie et al. [35] John et al. [1]	
Greece	Ionian Sea	Tsiamis et al. [36]	the Greek coasts
Hungary	freshwater	Mareš et al. [12]	
France	freshwater	Anon [37]	
Italy	marine	Sfriso [38]	Veneto, Mediterranean Sea
Montenegro	freshwater (a river, a ditch, the Milet Canal) and marine (on the rocky shore of the Adriatic Sea)	Czerwik-Marcinkowska et al.	
Poland	from freshwater to hyperhaline, and brine habitats	Mareš et al. [12] Messyasz et al. [18] Richter and Pietryka [19]	mats
Portugal	ponds	Favot et al. [39]	the Ria Formosa Lagoon
Slovakia	freshwater	Mareš et al. [12]	
Sweden		Mareš et al. [12]	
**North America**			
USA	freshwater	Lougheed and Stevenson [13]	Lake Michigan, Muskegon Lake
**South America**			
Argentina	freshwater	Boraso and Zaixso [40]

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
