# Peer review of "X-ray Fluorescence Techniques in Determining the Habitat Preferences of Species—Ulva pilifera (Ulvales, Chlorophyta) from Montenegro Case Study"

_molecules, 2020, doi:10.3390/molecules25215022_

Round 1

Reviewer 1 Report

I have now finally (apologies for the delay in reviewing your manuscript) thoroughly read and reviewed your article "X-ray Fluorescence Techniques in Determining the Habitat Preferences of Species - Ulva pilifera (Ulvales, Chlorophyta) from Montenegro Case Study" I have a number of major reservations about this paper which I will deal with below under "Major comments".

First of all, the way how the manuscript is written requires an extensive editing of English language and style. It is not clear along the manuscript which species was used and corrections of formatting are required throughout the text (e.g. font, tables, decimal places, species name). Please review throughout the manuscript.

Figures 1 and 2 (description/location of the study areas) should be mentioned in the material and methods section and not in the results and discussion section.

Figures 3 B and C are the same picture and the caption is not correct. Figure 2 shows the sampling areas and in Figure 3 the caption is not clear and the freshwater and marine sampling sites are not in accordance with what was mentioned in Figure 2.

For the analysis was used a composite sample or a unique specimen? If a unique specimen was used, how did you guaranty that it was a single individual? This should be described in the materials and methods.

Did you use any reference material? Why?

In section 2.3. Elemental analysis, it doesn’t make sense to use elements abbreviation in the first line and then write the name of the element with the associated abbreviation.

It would be interesting to do ANOVAs for the elemental analysis.

Sincerely

Author Response

Dear Doris You
Assistant Editor

I would like to thank you for the remarks about our manuscript "X-ray fFluorescence Techniques in Determinig the Habitat Preferences of Species – Ulva pilifera (Ulvales, Chlorophyta) from Montenegro Case Study". I am grateful to Reviewer for his comments. Below, I answered the address questions, and I am sending the correcting manuscript.

Yours sincerely,

Reviewers' comments:

Review of revision:

The manuscript must be revised by a native English speaker.

Thanks very much for your suggestion. I agree with a change to the editing of English language and style. I asked a specialized agency to proofread the entire manuscript. And I recived a certificate the proofreader applied the rules of British English.

All measurement units must be used in the same form in the entire manuscript (i.e., mg L-1 or mg/L).

As you suggested I added all measurement units in the same form in the entire manuscript.

In the Abstract it is mentioned that Ca=16.3 ppm, but in the Results Section (Table 2) the Ca content is 16.3%. The difference is important. Please revise the data!

I apologize for the confusion with Abstract and Results Section.

The samples were collected from 4 sites, but data were reported just for 3 sites because "site C was below the limit of significance". Please explain the limit of significance!

The uncertainty values are the expanded uncertainty with an expansion of level about 95% with factor of k = 2. The value of extended uncertainty is given as a range (± U95%).

In Section 3 it is mentioned that water samples were collected and analyzed for "nutrients, trace elements and metals", but the obtained data were not reported. Also, the correlations between algae content and water content are missing and so, the affirmation: "Ulva piliferacould be used as a bioindicator of trace elemental composition in water and sediments" is not sustained by the data.

Thank you for your suggestion. The Authors of manuscript are preparing the other manuscript and reported the correlations between algae and water content.

In Section 3, subsection 3.1. Study site, I recommend keeping just the sites description, and all procedure (i.e., cleaning, drying, optical microscopy) to be part of other subsections.

I introduced the suggested corrections.

I recommend inserting in Section 3 a subsection entitled "Samples preparation" in which will be described the sample preparation for each techniques used in this study.

Thanks very much for your suggestion. I agree with a inserting in Section 3 a subsection entitled: Samples preparation where I described the samples preparation for each techniques used in study.

The Reference list must be revised because some references are written with the underlined text (i.e. [1], [3], [19], [27], [30], [35], and [36]), or do not respect the format (i.e., [41-58]). 

Thanks for the remark, I introduced the correction of the references.

First of all, the way how the manuscript is written requires an extensive editing of English language and style. It is not clear along the manuscript which species was used and corrections of formatting are required throughout the text (e.g. font, tables, decimal places, species name). Please review throughout the manuscript.

Thanks very much for your suggestion. I agree with a change to the editing of English language and style. I asked a specialized agency to proofread the entire manuscript. And I recived a certificate the proofreader applied the rules of British English.

Figures 1 and 2 (description/location of the study areas) should be mentioned in the material and methods section and not in the results and discussion section.

I apologize for the confusion with different places Figures 1 and 2. I introduced this Figures in the section: the material and methods.  

Figures 3 B and C are the same picture and the caption is not correct. Figure 2 shows the sampling areas and in Figure 3 the caption is not clear and the freshwater and marine sampling sites are not in accordance with what was mentioned in Figure 2.

Thank you very much for Reviewer comment. I introduced correction of Figure 2 and Figure 3 where appropriate.

For the analysis was used a composite sample or a unique specimen? If a unique specimen was used, how did you guaranty that it was a single individual? This should be described in the materials and methods.

Thank you for your comments. A total of 20 Ulva samples were collected from four sites. Subsequently, each sample was washed clean with water and thoroughly dried with absorbent paper. Of each specimen, a piece of approximately 1 cm2 was preserved in silica. Each sterile clean bottles was labeled with the date of collected, the habitats number, and sample number. The remainder of each individual collected was preserved as herbarium specimens. This identiphication system allowed a visual comparison with herbarium specimens.

Did you use any reference material? Why?

Yes, the study was focused on individuals collected exclusively from different water ecosystems and carried out the taxonomic revision of Ulva taxa originating from herbarium specimens.

In section 2.3. Elemental analysis, it doesn’t make sense to use elements abbreviation in the first line and then write the name of the element with the associated abbreviation.

Thank you, I introduced the suggested corrections.

It would be interesting to do ANOVAs for the elemental analysis.

Thank you for your comments. The molecular methods used in the determination of elements allowed to determine their content by a single measurement for each element. The lack of repetitions meant that we could not use any of the simple statistical tests. However, a multidimensional character of the collected data allowed us to use ordination techniques, from which we chose the PCA indirect ordinance analysis and the CVA discriminant analysis. These analyzes allowed us to indicate regularities in the studied set and to determine the inter-site diversity.

Reviewer 2 Report

All suggestions were made directly on the manuscript, but also, they are listed below (generally):

  1. The manuscript must be revised by a native English speaker.
  2. All measurement units must be used in the same form in the entire manuscript (i.e., mg L-1 or mg/L).
  3. In the Abstract it is mentioned that Ca=16.3 ppm, but in the Results Section (Table 2) the Ca content is 16.3%. The difference is important. Please revise the data!
  4. The samples were collected from 4 sites, but data were reported just for 3 sites because "site C was below the limit of significance". Please explain the limit of significance!
  5. In Section 3 it is mentioned that water samples were collected and analyzed for "nutrients, trace elements and metals", but the obtained data were not reported. Also, the correlations between algae content and water content are missing and so, the affirmation: "Ulva pilifera could be used as a bioindicator of trace elemental composition in water and sediments" is not sustained by the data.
  6. In Section 3, subsection 3.1. Study site, I recommend keeping just the sites description, and all procedure (i.e., cleaning, drying, optical microscopy) to be part of other subsections.
  7. I recommend inserting in Section 3 a subsection entitled "Samples preparation" in which will be described the sample preparation for each techniques used in this study.
  8. The Reference list must be revised because some references are written with the underlined text (i.e. [1], [3], [19], [27], [30], [35], and [36]), or do not respect the format (i.e., [41-58]).  

Author Response

(The authors gave the same response as above.)

Round 2

Reviewer 2 Report

Few suggestions were done in .pdf (as comments!).

Please pay more attention to the measurement units, font size and reference list!

Author Response

Reviewer’s comments:

Ca=16.3% (erase the spaces)

Thank you very much for Reviewer comment. I introduced changes to the phrase.

[2-4]

I introduced a correct.

Unfortunately, in the comments done in .pdf I can not write the correct form of measurement units. I use the symbol "^" to mark that the value after it are written as superscript.

I have mention the correct form in the comments done for review 1 in the platform!

Thank you for your comments. Of course I made a mistake. I added a correction all measurement units in manuscript.

The letters used for figures numbering (a, b, c, and d) must be the same as they are used in the Figure 4 caption (A, B, C, and D). Please choose if you use capitalized letters or not and use that style on the entire manuscript!

Thank you for your comments. I introduced a change.

The concentrations values must be written using the same number of decimals (2 or 3). Also, this number of decimals must be used for measurement uncertainty.

I added missing number of decimals.

The concentrations values must be written using the same number of decimals (2 or 3). Also, this number of decimals must be used for measurement uncertainty.

In the Table header I do not see the "μ" symbol!

I added missing number of decimals and "μ" symbol in the Table header.

mg L-1 Cl-  

This is the correct form of measurement unit! Please revise the entire manuscript for the measurement units!

Thank you for your comments. I completed information as you suggested.

"Calcium" and "barium" have different font size and have hyperlink to wikipedia!

Please remove the hyperlink and write these words with the same font as the entire manuscript!

Thank you for your comments. I introduced a change.

This paragraph is written with different font sizes!

I introduced a change.

HNO3

I introduced a change.

Delete the space between value and °C.

As you suggested I deleted the space between value.

delete "at"

As you suggested I deleted the “at”.

According

I added the correction.

mm2

I added the correction.

font size!

I changed the font size in the phrase.

The reference list do not respect the requirements!

Many thanks for your comments. I introduced changes to the references.